# How Robustly Do LLMs Understand Execution Semantics?

Claudio Spiess
cvspiess@ucdavis.edu
UC Davis
Davis, California, USA

Prem Devanbu
pdevanbu@ucdavis.edu
UC Davis
Davis, California, USA

Earl T. Barr
e.barr@ucl.ac.uk
University College London
London, UK

## Abstract

LLMs demonstrate remarkable reasoning capabilities, yet whether they utilize internal world models or rely on sophisticated pattern matching remains open. We study LLMs through the lens of *robustness of their code understanding* using a standard program-output prediction task. Our results reveal a stark divergence in model behavior: while open-source reasoning models (DeepSeek-R1 family) maintain stable, albeit somewhat lower accuracies (38 % to 67 %) under code transformations & input perturbations, the frontier model GPT-5.2 exhibits surprising instruction-sensitive brittleness. Despite achieving a near-perfect score of 99 % on the original, unperturbed CRUXEVAL benchmark, perturbed inputs trigger accuracy declines between 20 % and 24 %. In addition, we find that many models perform much worse at predicting behavior on perturbed inputs that raise exceptions, *and* that prediction performance depends on the kind of exception. We study remedies to address this deficiency in exception prediction, and evaluate the effect of these remedies on the ability to predict non-exception behaviors. Our findings both point to limitations in the way all models understand code, and reinforce and extend the value of using perturbation to evaluate code models.

## CCS Concepts

• **Computing methodologies → Artificial intelligence**.

## Keywords

code understanding, robustness, metamorphic testing

**ACM Reference Format:**
Claudio Spiess, Prem Devanbu, and Earl T. Barr. 2026. How Robustly Do LLMs Understand Execution Semantics?. In *Proceedings of the 3rd ACM International Conference on AI-Powered Software (AIware '26), July 6–7, 2026, Montreal, QC, Canada.* ACM, New York, NY, USA, 11 pages. https://doi.org/10.1145/3805760.3814919

## 1 Introduction

Current LLMs show impressive proficiency in Software Engineering tasks, such as code generation [28], code summarization [43], and code repair [54]. This raises the question: to what extent does language model performance arise from actually *understanding* the semantics of code? Studies show that developers spend the majority of their time understanding existing code [49, 29]; the capacity for *understanding* code is thus vital to human coding work. For example, maintenance work can require understanding what a buggy program actually does, in response to a failure-triggering input. Thus, it is natural to ask if LLMs are similarly able to understand code.

Language model "understanding" is a topic well-explored for natural language, using benchmarks such as MMLU [21] and Super-GLUE [44]. Recently, several benchmarks have tackled this issue for code. CodeMMLU [34] comprises multiple-choice tests that evaluate several types of code understanding (syntax and semantics). CRUXEVAL [17] is more demanding, requiring LLMs to understand programs well-enough to *precisely predict exact outputs* for given test inputs (not just select one answer from presented choices, as with CodeMMLU).

To effectively maintain a program, one must understand it sufficiently well to *robustly* and consistently predict its behavior, regardless of how exactly it is coded, which inputs it is given, or the intricacy of their execution paths. Predicting outputs given inputs is a vital capability for software maintenance tasks: developers need to do this when diagnosing failures, when refactoring the code or adding enhancements. LLMs perform surprisingly well on this task, with frontier models predicting outputs for inputs for published benchmarks [17] with over 99% accuracy, but how **robust** is this performance? In other words, how much can a developer rely on LLM predictions while debugging or maintaining code?

Prior work suggests that language model understanding of code is *not* robust with respect to program structure for highly demanding tasks such as code equivalence and static analysis [23, 47]. In this paper, we aim to refine the benchmarks for evaluating LLM code understanding that leverage output prediction as a proxy for code understanding. We define robustness as the ability of a model to consistently predict a program's output correctly on a given input despite variations that do not alter the program's semantics. To measure it, our study contributes an evaluation of LLM robustness across three critical dimensions:

First, we consider robustness of LLMs on output prediction to *meaning-preserving transformations* (MPTs). In doing so, we provide a rigorous reproduction of prior work [45, 30], evaluating LLM performance on syntactically different, but semantically equivalent, program variants. This sets an empirical anchor, both for the community and for this work: it establishes the baseline of model stability required to contextualize the more brittle behaviors we uncover regarding input neighborhoods and execution traces. In RQ2, we enhance this reproduction by employing type-aware transformations, ensuring the validity of the generated variants.

> **RQ1:** How robust is LLM code understanding to program perturbation via meaning-preserving code transformations?

*RQ1 Finding:* Our results largely align with and extend the findings of prior work [45, 30], confirming that most models suffer modest decreases under meaning-preserving transformations. However, we identify notable exceptions to this trend in frontier models, where these transformations occasionally expose unexpected failures in execution reasoning (Section 4.1).

Next, we evaluate whether LLM output prediction is robust *under input perturbation*. We argue that true code understanding requires navigating more than a single "happy path". Thus, we extend output prediction as a lens for code understanding by moving beyond point-wise evaluation to consider input neighborhoods, generated by type-aware mutation. Unhappy paths include program failure; robust code understanding requires an LLM to to correctly and consistently predict behavior on failing inputs. We quantify model performance across this dimension of execution semantics, which has been neglected in the evaluation of code LLMs.

> **RQ2:** How robust is LLM code understanding to input perturbation? How well do LLMs predict runtime exceptions for perturbed inputs that cause exceptions?

*RQ2 Finding:* We find that LLMs frequently fail to make consistent (all right or all wrong) predictions even within these constrained input neighborhoods, both (unexpectedly) in some large, frontier models, *and* also with exception-raising inputs. This behavior undermines confidence in their underlying world models. Section 5 explores these issues in detail.

Finally, we move beyond surface-level input-output pairs to study the robustness of LLM output prediction at the level of execution traces. While prior work treats the program as a black box, we use dynamic analysis to correlate prediction accuracy with the *number of discrete control flow decisions* encountered during a run. This provides a granular, trace-based view of execution reasoning that has been previously overlooked. We investigate whether model robustness is sensitive to the density of these decisions, an objective proxy for the reasoning load required to reach a program's output.

> **RQ3:** How robust is LLM code understanding to control flow decisions? Are execution paths with more decisions, more difficult to reason about?

*RQ3 Finding:* Our data broadly suggests that LLMs are less accurate at predicting outputs as more decisions are encountered along an execution trace. When the decisions are primarily loop control conditions, as opposed to `if-else` chains, this finding points to disfluencies in LLM's understanding of iteration (Section 4.3).

## 2 Background

This section describes the CRUXEVAL benchmark, which we use in our evaluation. We then establish the performance and robustness metrics used throughout this study, including our novel measures for quantifying consistency across input neighborhoods.

### 2.1 CruxEval

The CRUXEVAL benchmark is designed to assess the code reasoning capabilities of LLMs on functions. Unlike standard code generation benchmarks, CRUXEVAL tests a model's ability to act as an interpreter and reason about program semantics. The benchmark consists of 800 distinct 2–13 line Python functions $\mathcal{P}$, where each $p \in \mathcal{P}$ is paired with a specific input $x_p$, which may be a tuple of arguments, and its corresponding return-value output $y_p$ recorded by execution.

On the output prediction task, a model is prompted with the function code and associated input, and must determine the result. For example, if given the function `def f(s): return s.upper() + "!"` and the input "hello", the model must correctly predict the execution output by completing the assertion `assert f("hello") == ??` with "HELLO!".

Each program is paired with a single input and nonerroneous output. We argue that this point-wise, happy path evaluation is insufficient to assess generalized code understanding of LLMs. Thus, we extend CRUXEVAL in two ways. First, we adapt it to consider sets of inputs; a model's ability to predict behavior across a range of values is essential for assessing whether its understanding is robust and establishing confidence in that finding. Second, we extend CRUXEVAL to include to type-invalid inputs and exception outputs.

### 2.2 Evaluation Metrics

To evaluate the robustness of LLMs on code execution tasks, we distinguish between standard performance and structural consistency. Given a set of programs $\mathcal{P}$, where each program $p \in \mathcal{P}$ is associated with the original input $x_{p,0}$ (from the original CRUXEVAL benchmark) and a set of $n$ synthetic variant inputs $\{x_{p,1}, \ldots, x_{p,n}\}$, we utilize five metrics.

We use three measures of accuracy:

**Accuracy $C_o$:** The model's accuracy on the original CRUXEVAL input set. It serves as the baseline for comparison with prior work, and is equivalent to Pass@1.

**Accuracy $C_{\tilde{o}}$:** The model's average accuracy across all perturbed input variants CRUXEVAL$_o$, excluding the original inputs.

**Accuracy $C_{o \cup \tilde{o}}$:** The mean accuracy across the union of original and perturbed inputs.

To quantify the reliability of these predictions, we introduce two additional measures.

**Program-level Strict Robustness (PSR)** is the fraction of programs where a model predicts the output correctly for all $n$ inputs in a neighborhood:

$$\text{PSR} = \frac{1}{|\mathcal{P}|} \sum_{p \in \mathcal{P}} \prod_{i=0}^{n} \mathbb{1}\left(f(p, x_{p,i}) \cong \hat{y}_{p,i}\right) \quad (1)$$

Unlike standard accuracy, PSR assesses whether a model's understanding is stable across a function's input space. While standard accuracy can be inflated by 'lucky' hits on some inputs, PSR requires a model to remain correct across a set of related inputs. By treating the program as the unit of success rather than the individual input, PSR acts as a diagnostic for decision consistency, distinguishing between robust execution reasoning and spurious correlations or local brittleness.

Equation (1) relies on the congruence relation $y \cong \hat{y}$ whose definition rests on task-specific semantics, rather than strict syntactic or interpretive equality. Under this relation, a predicted output $\hat{y}$ is congruent to the ground truth $y$ if it preserves the invariant properties required by the task — such as indicating the correct exception or branch decision — independent of its literal string representation. We discuss how we instantiate $\cong$ in Section 3.2, RQ2.

Finally, we define **Robust Drop ($\mathcal{R}_\Delta$)** as the performance difference between original and perturbed inputs:

$$\mathcal{R}_\Delta = \text{Accuracy}_{C_{\ddot{o}}} - \text{Accuracy}_{C_o} \qquad (2)$$

A high positive value suggests a model that relies on "happy path" memorization, while values near or below zero suggest robust execution reasoning that is invariant to (or even aided by) input perturbation.

## 3 Methodology

This section presents our methodology, starting with our benchmark and model selection, and closing with our experimental design for each RQ.

### 3.1 Benchmark and Model Selection

We centered our study on Python, because of its industrial importance, and the CRUXEVAL benchmark, because its ubiquity in prior work facilitates cross study comparison, and because it provides a stable basis for our experimental pivot: testing the robustness of LLM code understanding when the single "happy path", or the 'single input-output', constraint is removed. By isolating this variable within the most recognized code execution reasoning benchmark, we have uncovered a instruction-sensitive brittleness in current LLM execution reasoning that sheds new light on prior results.

To provide a view of how LLM architecture influences code understanding, we selected a diverse suite of 14 models. Our selection criteria prioritized model size, training regime, and access model (open-weights *vs.* proprietary). The resulting cohort includes: DeepSeek-R1-Distill 1.5B, 7B, 8B [13], Llama 3.1 8B [16], Qwen2.5 Math 1.5B and 7B [53], Qwen2.5 Base and Instruct (7B and 14B) [37], NVIDIA OpenCodeReasoning Nemotron 7B and 14B [1], GPT-5 Nano, GPT-5.2, and Gemini 3 Pro. We cover a parameter size range from 1.5B to 14B (proprietary model size unknown at time of writing), four training regimes (completion, instruction-following, reasoning, and code-reasoning), and differing access models (3 proprietary, 11 open-weights).

DeepSeek-R1-Distill models[1] are fine-tunes ("*distillations*") of existing open-weight "base" models, using generations from the 671B parameter DeepSeek-R1 model trained with reasoning GRPO [39]. GRPO is a reinforcement learning algorithm utilized in R1 training to instill extended chain-of-thought reasoning by evaluating and scoring groups of outputs relative to one another, eliminating the need for a separate critic model. To isolate the effect of GRPO reasoning training over prior training approaches, our study includes both the R1-Distill models and their respective base model counterparts. For example, Qwen2.5-Math 1.5B and 7B are the base, starting checkpoints of DeepSeek-R1-Distill 1.5B and 7B.

For inference of open-weight models, we utilized HuggingFace `text-generation-inference`[2] (3.3.6) on 2 x TITAN RTX GPUs. We utilized greedy decoding (*i.e.*, temperature=0), and a maximum generation length of 16,384. To follow developer workflows using *frontier* models, which often neglect manual parameter adjustment [14], we accessed GPT-5.2, GPT-5 NANO, and GEMINI-3-PRO via their default API configurations. We note that GEMINI-3-PRO uses dynamic (high) thinking level by default, whereas GPT defaults to medium. To mitigate experimental bias, we performed each query independently to avoid context-leakage [18]. No explicit system prompts were utilized, ensuring the evaluation focused strictly on the models' baseline performance and inherent knowledge [33].

### 3.2 Experimental Design

We now detail our experimental design for each of our RQ.

*RQ1.* We perturb the original CRUXEVAL$_o$ programs with meaning-preserving transformations (MPTs) to produce two variants. The first is CRUXEVAL$_p$, built using four syntactic transformations, detailed in *Syntax Transform* below. The second is CRUXEVAL$_v$, built via variable renaming detailed below. We execute the code with the original input and verify output equivalence as a sanity-check of semantic preservation.

We hypothesize that simultaneously applying MPTs and input perturbations would create antagonistic effects, further degrading LLM performance on the CRUXEVAL output prediction task. To help characterize these dynamics, we isolate the individual contributions of each class of perturbations in this paper. This granular approach is a prerequisite for understanding the underlying failure modes before addressing the compounding complexities of their interactions, which we reserve for future inquiry.

**Syntax Transform** Following Chakraborty et al. [9], we apply OperandSwap, BlockSwap, ForWhileSwap, and DeadCodeInsertion transformations, producing CRUXEVAL$_p$. Only a single transformation was applied per program. Thus, each program has one transformed counterpart, except for 88 programs where no valid transformation was produced. Upon examination, these were predominantly very short programs without control flow.

**Variable Renaming** We mine DYPYBENCH, a dataset of 50 large, popular Python projects [7], for identifiers and their occurrences. For each CRUXEVAL program, we rename all identifiers with randomly sampled ones, weighted by number of occurrences, from the mined identifier distribution, *i.e.*, the most common identifiers are more likely to be sampled. This ensures that the identifiers are unlikely to be rare, and thus carry higher likelihood for the model, while also likely to be out-of-context. Thus, the renamed variables are generic, and unlikely to be relevant; we refer to this setting as CRUXEVAL$_v$.

Our evaluation begins with the original CRUXEVAL few-shot prompt [17], as shown in Figure 2. The answers are in the form `[ANSWER] assert $expr$ == $value$ [/ANSWER]`. We extract the predicted result $value$ from model responses using heuristics and regular expressions, ensuring that left-hand side of the assertion matches the query ground truth. We compare the extracted output

---

(right-hand side) with the gold-truth output recorded during execution using a Python expression evaluator[3], minimizing the effects of stylistic differences, such as quotations and whitespace.

We found that reasoning models frequently produced conversational, natural language explanations, chain-of-thought, or further examples using the provided function, *after* the correct answer. The original CRUXEVAL evaluation harness assumes plain auto-regressive completion, and only considers the last answer. To not penalize models in such cases, we label a response containing multiple assertions as correct, if *at least one* extracted assertion is correct, *i.e.*, both the left and right-hand side of the assertion match the ground truth. This relaxation of the answer-extraction protocol accommodates chat-style reasoning models that fail the standard CRUXEVAL harness, while avoiding confounding factors any changes in the prompt would introduce.

*RQ2.* We perturb CRUXEVAL inputs using a type-aware mutation algorithm following Liu et al. [32]. We produce perturbed inputs using the original input & program pairs in CRUXEVAL. We mutate the original CRUXEVAL input, reject the input if it is already in the set of inputs for that particular program, and otherwise execute the code with that input, record the output, and save samples that raise exceptions separately in a new CRUXEVAL$_{exc}$ dataset. If an exception is raised, we record its type and message *e.g.*, IndexError and 'list index out of range'.

As CRUXEVAL programs are small and simple, we enforce a 5 second execution time limit during this perturbation phase. We note that the evaluation phase remains unaffected, since the function does not need to be executed again. As in the original CRUXEVAL harness, a TimeoutError exception is raised when the time limit is exceeded, suggesting an infinite loop. Gu et al. [17] used a 3 s timeout. To account for instrumentation overhead and additional execution, we relaxed the timeout to 5 s to limit false negative risk *i.e.*, incorrectly marking valid, mutated input as invalid (exception raising).

We observed a maximum execution time of 7.95 ms with instrumentation overhead, and 976 μs for just the program-input excluding overhead. This suggests that the execution time limit has no effect on our non-exception results. We experimented with a doubled 10 s limit for CRUXEVAL$_{exc}$, and observed a maximum execution time of 4.09 ms for non-TimeoutError cases, and no change in outcome for the TimeoutError cases compared to the 5 s limit.

We limit the number of distinct perturbed inputs to 10 due to experimental (LLM usage) costs. Some programs ($n = 116$) have fewer than 10 inputs due to implicit input constraints *e.g.*, a program may enforce acceptable values for some argument $x$ with a cardinality < 10. We exclude such programs from our study to ensure equal sample sizes, and leave constraint-aware perturbation to future work. We repeat our model evaluation as described in RQ1.

The ability to predict that an exception will be raised while executing a given program-input pair $(p, x_{p,i})$, and specifically *which* exception, is helpful while debugging & maintaining code. CRUXEVAL ignores this issue; we, however, also gather & include input samples that raise runtime exceptions in our enhanced CRUXEVAL$_{exc}$, thus extending output prediction to cover exceptions.

[3]https://pypi.org/project/asteval/

**Table 1: Average model accuracy on canonical CRUXEVAL, code-perturbed CRUXEVAL$_p$, and CRUXEVAL$_v$. Asterisks \*, \*\*, and \*\*\* represent McNemar test $p < 0.05$, $0.01$, and $0.001$ respectively, for original *vs.* code-perturbed. As each model is a different experiment, we do not make claims across all models based the *p*-values and thus do not apply corrections.**

| Model | CRUXEVAL | CRUXEVAL$_p$ | CRUXEVAL$_v$ |
|---|---|---|---|
| Qwen2.5 Math 1.5B | 12.25 | 9.27* | 11.13 |
| DS R1 1.5B | 40.00 | 40.45 | 35.50* |
| Qwen2.5 Math 7B | 37.00 | 35.53 | 34.38 |
| Qwen2.5 Instr 7B | 42.63 | 42.98 | 40.50 |
| Nemotron 7B | 47.00 | 37.22*** | 44.75 |
| DS R1 7B | 62.63 | 60.25 | 61.50 |
| Llama 3.1 8B | 37.88 | 34.55 | 34.63* |
| DS R1 8B | 66.13 | 61.66 | 61.25** |
| Qwen2.5 14B | 22.50 | 27.53 | 16.00*** |
| Qwen2.5 Instr 14B | 48.88 | 45.65 | 47.00 |
| Nemotron 14B | 68.63 | 57.87*** | 66.13 |
| GPT-5 Nano | 97.20 | 87.92*** | 93.88*** |
| GPT-5.2 | **99.24** | 76.40*** | 75.75*** |
| Gemini 3 Pro | **99.24** | 95.79*** | **98.37** |

For exception prediction, we initially labeled a response as correct if the ground-truth exception type or message appeared anywhere in model output. However, review revealed cases where the model correctly predicted an exception, but failed to output either the exception type or its message verbatim. To minimize false negatives, we therefore devised heuristics for TimeoutError, ValueError, TypeError, and IndexError. For each, we defined 1-5 synonyms *e.g.*, 'infinite' for TimeoutError.

We then realized $\cong$ in Equation (1) in two ways: (1) **Partial Match**: The model's response contains an exception type or synonym, regardless of whether that matches the ground truth exception. This captures the ability to predict that *some* exception will occur. (2) **Exact Match**: The model's response contains the ground-truth exception type, synonym, or message. In contrast to (1), this captures the ability to predict *which* exception will occur.

*RQ3.* We employ both static analysis to identify programs with loop constructs, and dynamic analysis (*tracing*) to record execution behavior for a given input *e.g.*, executed line-numbers, number of loop iterations, *etc.* We focus on loops, *i.e.*, for/while and comprehensions. We record the number of decisions encountered in each loop *e.g.*, a for loop with one if, with a compound condition *e.g.*, (x > 5) and (y < 3) in the header (and no further conditions in its body), would have a total of **three** decisions: the iterator itself, and the two sub-expressions in the if condition. With two iterations, the execution encounters a total of *six* decisions.

## 4 Results

We report experimental findings for our three research questions, measuring robustness to code perturbation (RQ1), input perturbation (RQ2), and execution decisions (RQ3).

### 4.1 RQ1: Robustness to Code Perturbation

In Table 1, we observe generally minor differences when shifting from the original CRUXEVAL programs to meaning-preserving

**Table 2: Accuracy & robustness metrics for original *vs.* perturbed CRUXEVAL *inputs*. Arrow ↑ indicates higher values are preferred. Each model was evaluated on 684 programs, for which the original benchmark input and 10 unique perturbed variants were available.**

| Model | $C_o$ (↑) | $C_{\tilde{o}}$ (↑) | $C_{o \cup \tilde{o}}$ (↑) | PSR (↑) | $\mathcal{R}_\Delta$ (↑) |
|---|---|---|---|---|---|
| Qwen2.5 Math 1.5B | 12.48 | 12.22 | 12.24 | 1.91 | −0.26 |
| DS R1 1.5B | 38.45 | 37.50 | 37.59 | 4.39 | −0.95 |
| Qwen2.5 Math 7B | 34.94 | 33.27 | 33.43 | 11.11 | −1.67 |
| Qwen2.5 Instr 7B | 40.94 | 41.81 | 41.73 | 16.81 | +0.88 |
| Nemotron 7B | 46.20 | 43.26 | 43.53 | 4.39 | −2.94 |
| DS R1 7B | 65.06 | 62.62 | 62.84 | 27.34 | −2.44 |
| Llama 3.1 8B | 36.40 | 33.82 | 34.05 | 11.84 | −2.59 |
| DS R1 8B | 64.91 | 62.91 | 63.09 | 29.68 | −2.00 |
| Qwen2.5 14B | 22.08 | 23.20 | 23.10 | 3.80 | +1.13 |
| Qwen2.5 Instr 14B | 47.21 | 47.84 | 47.79 | 21.11 | +0.63 |
| Nemotron 14B | 67.25 | 65.06 | 65.26 | 33.48 | −2.19 |
| GPT-5 Nano | 96.78 | 96.61 | 96.62 | 82.75 | −0.18 |
| GPT-5.2 | **99.12** | 79.61 | 81.38 | 56.14 | −19.52 |
| Gemini 3 Pro | 99.10 | **99.36** | **99.34** | 97.16 | **+0.25** |

transformation variants, under both syntactic transformation and variable renaming (Section 3.2). In contrast to Section 4.2, we only perturb the program *code* in this setting, as opposed to the *inputs i.e.*, test cases. We note that discrepancies between Table 2 and Table 1 arise due to minor differences in program sets (712 *vs.* 684), as described in Section 3.2 RQ1 and RQ2.

Similar to our observations in Table 2, the reasoning distilled models show remarkable gains over their base counterparts. For instance, Llama 3.1 8B accuracy on CRUXEVAL jumps from 37.23 % to 67.82 % in its DeepSeek-R1-Distill ("DS R1 8B" in Table 1) version. However, these models also degrade under perturbation, with DeepSeek-R1-Distill 8B dropping to 61.66 % on MPT (−6.16 Δ) and 61.25 % (−6.57 Δ) on Renamed.

The impact of variable renaming and MPTs is particularly evident for the frontier model GPT-5.2, showing a sharp decline from a near-perfect 99.24 % to around 76.40 % for both MPT and Renamed. In contrast, the decline for GEMINI-3-PRO is much smaller, dropping from 99.24 % to 95.79 % and 98.37 % for MPT and Renamed, respectively. On our dataset, despite matching GEMINI-3-PRO's performance on the original CRUXEVAL, GPT-5.2 is more sensitive to surface-level syntactic code changes than GEMINI-3-PRO. This is surprising as GPT-5.2 is near-perfect on the original benchmark, is newer (Jan 2025 *vs.* Dec 2025), and likely larger in terms of parameter count. GPT-5 NANO, while also being a strong performer on the original code, is significantly impacted by MPT and Renaming although not as much as GPT-5.2 *per se*. These findings closely align with those reported by Lam et al. [30].

> **Finding RQ1:** LLM code reasoning is generally robust to MPT; performance drops from transformations or renaming are typically marginal (averaging -4.3% $\mathcal{R}_\Delta$); GPT-5.2 is the stark exception with a dramatic ≈−23 loss on both MPTs.

**Table 3: Accuracy of frontier models on predicting the precise exception that actually occurs ("Exact Match") or that some exception occurs ("Partial Match").**

| Model | Exact Match % | Partial Match % |
|---|---|---|
| GPT-5 Nano | **71.95** | **73.16** |
| GPT-5.2 | 14.77 | 14.77 |
| Gemini 3 Pro | 48.56 | 50.72 |

## 4.2 RQ2: Robustness to Input Perturbation

In Table 2, we observe that accuracy ranges widely across models, from as little as 12.48 %, to as high as 99.12 %. Our results show the effectiveness of reasoning post-training *e.g.*, QWEN2.5-MATH-1.5B's accuracy jumps from 12.48 % to 38.45 % after the DEEPSEEK-R1-DISTILL finetune. Our best open-source model reaches 67.25 % accuracy, far lower than the best-performing frontier model GPT-5.2's accuracy of 99.12 %.

Considering $\mathcal{R}_\Delta$, we observe four models that achieved higher accuracy on the perturbed inputs than the original input, resulting in positive $\mathcal{R}_\Delta$ values. On the other hand, the majority of open-weight models did worse on the perturbed inputs, with an average $\mathcal{R}_\Delta$ of −1.127. In contrast, the frontier model GPT-5.2 suffered a significant $\mathcal{R}_\Delta$ of −19.52.

We also report (PSR) numbers, the proportion of programs for which models predict *all outputs correctly*. These are much lower, reaching only 56 % for GPT-5.2. Arguably, correct performance for *several inputs for the same program* is a better measure of how well a model actually understands a program; if a model truly understands a program, it should get the output correct on all inputs! The drop for GPT-5.2 from 99 % overall correct *for all inputs* to just 56 % correct *for all programs*, suggests that its output prediction capability for a random program on *any* given input is much less reliable than its 99 % performance on the original CRUXEVAL might indicate. We explore this concerning issue further in Section 5.

A surprising performance gap exists between GPT-5 NANO and GPT-5.2. While GPT-5 NANO is marketed as a smaller, more efficient version of GPT-5 released in August 2025, and GPT-5.2 is marketed as a flagship model released in December 2025, GPT-5 NANO outperforms GPT-5.2 on $C_{\tilde{o}}$ by 17 %, $C_{o \cup \tilde{o}}$ by 15.24 %, and PSR by 26.61 %. Furthermore, its $\mathcal{R}_\Delta$ of −0.18 is greater than the $\mathcal{R}_\Delta$ of −19.52 for GPT-5.2. These results suggest that GPT-5 NANO is more robust than its nominally stronger counterpart. We hypothesize that it is a quantized, and possibly distilled, version of GPT-5. Prior work [4, 27, 6] suggests quantization exerts a regularizing effect, steering optimization toward flatter minima that exhibit greater robustness to perturbation, and a reduced tendency for overfitting. If our hypothesis holds, quantization-regularization may help explain the greater generalization we observe for GPT-5 NANO.

Notably, GEMINI-3-PRO exhibits very high performance and impressive robustness: its $\mathcal{R}_\Delta$ is small and *positive*, and its PSR on the perturbed datasets is not that much lower than its performance on original CRUXEVAL dataset.

We subsequently investigated, for the best-performing frontier models, the LLM's ability to reason about input perturbations that result in runtime exceptions. In Table 3, we report the frontier

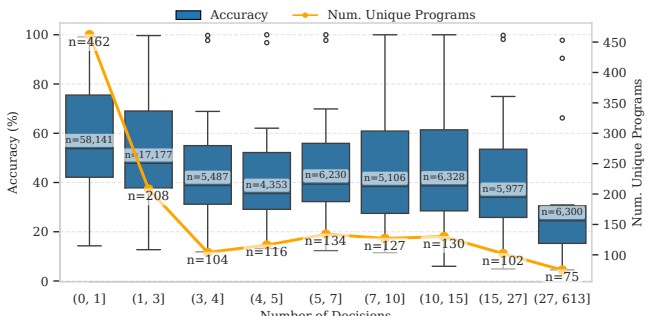

**Figure 1: Number of decisions encountered on execution path *vs.* model accuracy on output prediction. The yellow line shows the varying number of sample programs with increasing numbers of conditions on the execution paths**

model performances on CRUXEVAL$_{exc}$ using the canonical CRUXE-VAL prompt, for both correctness criteria. In general, we observe that models find it more difficult to predict exceptions. Table 3 shows the accuracy of our 3 best-performing frontier models on predicting exceptions; the first column is exact match, evaluating if the model can predict precisely which exception is thrown; the second column is "partial", which credits the model when any exception is predicted. It's noteworthy that models actually manage to predict the *right* exception in most cases that they manage to correctly predict that *some* exception is thrown. In Section 5, we explore the type of exceptions that are strictly correctly predicted, and then delve into the factors potentially related to poor exception performance, including the potentially prompt-compliant "sychophantic" tendency of instruction-tuned LLMs to produce a plausible, fake, non-erroneous output, even when the actual output is an error.

> **Finding RQ2:** Most models perform worse on perturbed inputs (average $\mathcal{R}_\Delta$ of −1.1), with GPT-5.2 being the most affected ($\mathcal{R}_\Delta$ of −19.5). Model performance noticeably degrades when predicting exceptions; we explore this in Section 5.

## 4.3 RQ3: Robustness to Control Decisions

We examine the results of decisions encountered during execution. In Figure 1, we visualize the results from Table 4. We investigate the relationship between the number of decisions encountered on an execution path for a given program-input pair $(p, x_{p,i})$, and model accuracy. Each box in the figure represents the 14 model accuracies across $n$ samples of output predictions for program-input pairs.

Due to the long-tail distribution of decision counts that arises from certain program-input pairs having many loop iterations, we bin the decision counts into nine bins for sufficient sample sizes—one for no decisions encountered (*e.g.*, straight line code), and eight *quantile* bins, with roughly comparable sample sizes varying from 4,353 to 17,177 observations.

Additionally, we plot the number of unique programs in each bin on the second (right) $y$-axis. Given that the sum of unique programs (1,458) exceeds the number of programs in CRUXEVAL, it follows

**Table 4: Binned number of decisions encountered during execution *vs.* mean model accuracy. Columns report number of samples (model, program, input triples), unique programs and unique inputs per bin.**

| #Decisions | #Samples | ACC (%) | #Programs | #Inputs |
|---|---|---|---|---|
| (0, 1] | 58,141 | 58.54 | 462 | 4,154 |
| (1, 3] | 17,177 | 55.48 | 208 | 1,227 |
| (3, 4] | 5,487 | 47.49 | 104 | 392 |
| (4, 5] | 4,353 | 45.16 | 116 | 311 |
| (5, 7] | 6,230 | 48.06 | 134 | 445 |
| (7, 10] | 5,106 | 48.45 | 127 | 365 |
| (10, 15] | 6,328 | 48.42 | 130 | 452 |
| (15, 27] | 5,977 | 44.75 | 102 | 427 |
| (27, 613] | 6,300 | 33.65 | 75 | 450 |

that execution paths and thus decision counts vary between inputs for the same program.

We observe a negative relationship between number of decisions and model output prediction correctness. A Mann-Whitney $U$ test on pooled model results indicated that the number of decisions encountered was significantly lower for correct predictions than for incorrect ones ($U = 1.436 \times 10^9$, $p < .001$). While the effect size as measured by the rank-biserial correlation was weak ($r_{rb} = −0.129$), this suggests that, as the number of decisions a model must reason about increases *e.g.*, an `if` statement in a `for` loop, its ability to correctly reason about the code execution decreases. Further analysis per model found a minimum $r_{rb}$ of −0.314 (NEMOTRON-14B), a median of −0.167, and maximum of 0.273 (GEMINI-3-PRO). As the only model with a positive (surprisingly strong) effect, GEMINI may be less hindered by the number of decisions.

Despite the overall negative trend in Figure 1, we note an increase in accuracy in the (5, 15] interval, where accuracy seems to recover somewhat from that in (3, 5]. The samples spike at this interval; further examination of this phenomenon and the GEMINI-3-PRO outlier is left to future work.

> **Finding RQ3:** Models generally struggle to predict outcomes as the number decisions encountered increases, with a significant drop-off at 3. GEMINI-3-PRO is the notable exception.

## 5 Discussion

We begin by noting the dramatically poor performance of frontier models in predicting outputs for CRUXEVAL$_{exc}$ compared to the non-exception-raising input set *e.g.*, GPT-5.2 accuracy drops from ~81% in Table 2, to ~15% for exception prediction in Table 3 (also reproduced in the "Original" line in Table 7).

Table 5 breaks down the accuracy of the models in predicting exceptions, based on the type of exception actually raised when running the code sample. Due to the small sample sizes for several exception types in CRUXEVAL$_{exc}$, the zero and 100 % accuracy results in Table 5 lack a reliable basis for comparison and are therefore excluded from this analysis.

The most commonly thrown exception is `TypeError`, with 358 occurrences; the least common is `NameError`, with just 3 occurrences. Each exception is a standard Python exception, with the exception of `TimeoutError`, which occurs when the code took

**Table 5: Distribution of exception types in CRUXEVAL_exc, along with model performance using the original CRUXEVAL prompt.**

| Exception | Count | Partial Match % (↑) | | | Exact Match % (↑) | | |
|---|---|---|---|---|---|---|---|
| | | Nano[a] | 5.2[b] | Gem3[c] | Nano[a] | 5.2[b] | Gem3[c] |
| TypeError | 358 | 58.82 | 5.32 | 31.56 | 56.02 | 5.32 | 27.65 |
| ValueError | 152 | 81.58 | 15.13 | 75.66 | 81.58 | 15.13 | 75.66 |
| IndexError | 138 | 95.49 | 24.64 | 62.32 | 95.49 | 24.64 | 61.59 |
| AttributeError | 73 | 94.44 | 1.37 | 26.03 | 94.44 | 1.37 | 26.03 |
| KeyError | 54 | 94.44 | 83.33 | 98.15 | 94.44 | 83.33 | 98.15 |
| TimeoutError | 36 | 27.78 | 0.00 | 63.89 | 27.78 | 0.00 | 63.89 |
| RecursionError | 10 | 100.00 | 0.00 | 100.00 | 100.00 | 0.00 | 70.00 |
| LookupError | 5 | 0.00 | 0.00 | 0.00 | 0.00 | 0.00 | 0.00 |
| ZeroDivisionError | 5 | 100.00 | 20.00 | 80.00 | 100.00 | 20.00 | 80.00 |
| NameError | 3 | 0.00 | 0.00 | 0.00 | 0.00 | 0.00 | 0.00 |

[a] GPT-5 Nano    [b] GPT-5.2    [c] Gemini 3 Pro

more than 5 seconds (wall time) to execute (see § 3.2), suggesting an infinite loop. We manually verified that the 36 occurrences, consisting of 11 programs, were indeed infinite `while` loops.

For each exception, Table 5 shows, for each model, when that particular exception is predicted precisely ("Exact Match"), and when the model predicts that *some* exception is predicted ("Partial Match"). An exact match may occur if one of the heuristic substrings for that *particular* exception is contained in the model output *e.g.*, 'infinite' for the ground-truth exception TimeoutError. Unlike infinite recursions (RecursionError), infinite loops are not a runtime exception in Python; thus, predicting the non-standard TimeoutError verbatim is challenging. We found that all correct predictions here are due to 'infinite' appearing in the model output, per the heuristic. We later find that with prompt variations, GEMINI-3-PRO can indeed predict TimeoutError *verbatim* for up to 19 % of cases.

In light of these two correctness criteria, we find that the performance gap is particularly evident in the 358 occurrences of TypeError exceptions: GEMINI-3-PRO correctly predicts that TypeError exception *per se* occurs just ~28% of the time, while predicting *some* exception ~32% of the time. In most instances of TypeError where GPT-5 NANO predicted *some* exception, it also almost always correctly predicted the specific TypeError, as evidenced by the minimal difference of 2.8 % between 'partial' and 'exact' match, 58.8 % and 56.0 % respectively. Upon further examination, we found that GPT-5 NANO correctly predicted the *exact* exception message for 48.74 % of TypeErrors.

Among the exceptions that occur more frequently, TypeError appears to be hardest one for all our frontier models[4]. In Python, TypeError functions as a catch-all, spanning, by convention, a diverse range of errors, such as argument count mismatches and non-iterable access. Python experts also use it as a broad, domain-extensible error type. This diversity likely accounts for both its prevalence in CRUXEVAL_exc and its relative prediction difficulty. Whether encountered statically or dynamically, this difficulty in

---

[4]While performance for LookupError & NameError are worse, we have too few samples of them to draw any conclusions.

**Table 7: Results for frontier models on CRUXEVAL_exc. *Exact Match* requires the generation to contain either the ground-truth exception type, exception message, or a heuristic substring. *Partial Match* requires the generation to contain *any* exception type or heuristic substring *e.g.*, model output contains "infinite", but ground-truth exception is TypeError.**

| Model | Prompt | Exact Match % | Partial Match % |
|---|---|---|---|
| GPT-5 Nano | Original | 71.95 | 73.16 |
| | Exceptions | 88.49 | 93.65 |
| | Type | 73.50 | 74.70 |
| | Exc. + Type | **89.93** | **95.92** |
| GPT-5.2 | Original | 14.77 | 14.77 |
| | Exceptions | 80.58 | 88.61 |
| | Type | 15.59 | 15.71 |
| | Exc. + Type | **81.30** | **91.13** |
| Gemini 3 Pro | Original | 48.56 | 50.72 |
| | Exceptions | 98.20 | 98.92 |
| | Type | 81.89 | 84.41 |
| | Exc. + Type | **98.92** | **99.64** |

predicting TypeErrors cannot be ignored: TypeErrors *will be* encountered when writing & debugging code; thus, it would be desirable to improve the ability to predict occurrences of this error. We begin by first discussing possible reasons of poor performance in predicting exceptions in general, and then turn more specifically to TypeErrors.

First, we note that the ~15% value in "Original" line for GPT-5.2, in Table 7 was produced using the actual, unmodified CRUXEVAL prompt [17]. Notably, this original prompt doesn't mention exceptions, as you can see if you inspect the CRUXEVAL prompt, labeled "Direct Output Prompt" in Figure 2. This lacuna may have caused the frontier LLMs to perform badly. Instruction-tuned models have been reported to show sycophancy [40], where they follow instructions literally & narrowly, ignoring other relevant context. For further clarity, we modified the CRUXEVAL prompt, adding **just** the blue text in the "Direct Output Prompt Type-Strict + Exceptions" prompt in the lower part of Figure 2, to tell the frontier LLMs to consider exceptions. This text comprises both an instruction and adds a few-shot example.

The results, as shown on the "Exceptions" line in Table 7, improve dramatically from ~15% for GPT-5.2 to ~81%, which is in the range of what we note for non-exception-raising inputs; in fact, it is able predict the presence of *some* exception ~88% of the time. Performance also improves for GEMINI-3-PRO, from ~49% to ~98%, and for GPT-5 NANO, from ~72% to ~88%.

We now focus on the high prevalence of Type-related exceptions in Table 5. We hypothesized that models might be too narrowly focused on predicting an output *value*, and ignoring *types*; this seems specially undesirable for a dynamically-typed language like Python, where the interpreter tracks *both types and values*. We therefore tried additional prompting **just** for type-tracking, shown in the "DOP Type-Strict + Exceptions" prompt in red text. We find that type-track prompting does improve performance, although never as much as exception-track prompting; it helped a lot with Gemini, but less so for the two GPT models ("Type" line, Table 7). Delving into the data, we find that the type-track prompting doesn't

**Figure 2: The prompts we used: the "Direct Output Prompt" is from the original Crux​Eval paper [17]. The lower prompt adds instructions for *both* type-tracking and exception-tracking. We also experimented with each separately**

---

**Direct Output Prompt**

```
You are given a Python function and an assertion containing an input to the
function. Complete the assertion with a literal (no unsimplified expressions,
no function calls) containing the output when executing the provided code on
the given input, even if the function is incorrect or incomplete. Do NOT output
any extra information. Provide the full assertion with the correct output in
[ANSWER] and [/ANSWER] tags, following the examples.
[PYTHON] def f(n): return n assert f(17) == ?? [/PYTHON] [ANSWER] assert f(17)
== 17 [/ANSWER]
[PYTHON] def f(s): return s + "a" assert f("x9j") == ?? [/PYTHON] [ANSWER]
assert f("x9j") == "x9ja" [/ANSWER]

[PYTHON]  {code}  assert f( {input} ) == ?? [/PYTHON] [ANSWER]
```

---

**Direct Output Prompt (Type-Strict + Exceptions)**

```
You are given a Python function and an assertion containing an input to the
function. Complete the assertion with a literal (no unsimplified expressions,
no function calls) containing the output when executing the provided code on
the given input. STRICTLY ANALYZE both the TYPES and VALUES of every variable.
Consider how type-specific operations and implicit type conversions affect
the final result. If the code raises an exception, complete the assertion with
the exception type and message. Do NOT output any extra information. Provide
the full assertion with the correct output in [ANSWER] and [/ANSWER] tags,
following the examples.
[PYTHON] def f(n): return n assert f(17) == ?? [/PYTHON] [ANSWER] assert f(17)
== 17 [/ANSWER]
[PYTHON] def f(n, p): return n[p] assert f([1, 2, 3], 4) == ?? [/PYTHON]
[ANSWER] assert f([1, 2, 3], 4) == "IndexError: list index out of range"
[/ANSWER]

[PYTHON]  {code}  assert f( {input} ) == ?? [/PYTHON] [ANSWER]
```

---

*specifically help improve* TypeError *identification a great deal*; in fact, for the GPT models, it helps ValueError identification more! Interestingly, the Exception track prompting helps find TypeErrors *much more* than Type-tracking, in all models.

We also tried combining type-track prompting with exception-track prompting: this is the "DOP Type-Strict + Exceptions" prompt including both the blue and red additions. As seen in the "Exc. + Type" line, their combination always works best, although the improvements over the "Exceptions"-only prompt are modest.

While the combination prompt improves performance *for all* the frontier models, it does raise the question as to whether this improvement is only manifest for the exception-raising cases: would the combination prompt actually help for normally-running inputs as well? To study this, we used the combination prompt on all our normally-running inputs (both the original *and* perturbed inputs). The results are in Table 8.

For both Gpt-5 Nano and Gemini-3-Pro, when we compare the change from "Original" prompt to the combined "Exc. + Type" prompt, we see improved performance for the original ($C_o$), the perturbed ($C_{\tilde{o}}$) and both together ($C_{o \cup \tilde{o}}$); we also see improved performance on the PSR score (the fraction of programs where every prediction is correct). Not surprisingly, improvements are larger for Gpt-5 Nano than for Gemini-3-Pro, which already performed almost perfectly with the Original prompt. We do notice a substantial performance drop-off in Gpt-5.2 when we use the combination prompt, from ∼81% to ∼76%. To investigate this 5% drop in accuracy, we inspected the results to check if Gpt-5.2 is sycophantically

**Table 8: Evaluating the combined Type & Exception Tracking prompt on the (original and perturbed) inputs that run without exceptions**

| Model | Prompt | $C_o$ (↑) | $C_{\tilde{o}}$ (↑) | $C_{o \cup \tilde{o}}$ (↑) | PSR (↑) |
|---|---|---|---|---|---|
| GPT-5 Nano | Original | 96.78 | 96.61 | 96.62 | 82.75 |
| GPT-5.2 | Original | **99.12** | 79.61 | 81.38 | 56.14 |
| Gemini 3 Pro | Original | 99.10 | **99.36** | **99.34** | **97.16** |
| GPT-5 Nano | Exc.+Type | 97.08 | 96.87 | 96.89 | 83.63 |
| GPT-5.2 | Exc.+Type | 76.17 | 75.98 | 76.00 | 50.29 |
| Gemini 3 Pro | Exc.+Type | **99.42** | **99.33** | **99.34** | **97.22** |

predicting more exceptions in response to the combination prompt. We found that in about ∼4% of the cases where it originally was correctly predicting the right values, it was *indeed* erroneously predicting exceptions. The causes for the rest of the errors remain unclear. As noted earlier in Section 4.2, we hypothesize the worse behavior of Gpt-5.2 relative to Gpt-5 Nano may be due to the latter being better regularized.

## 5.1 Threats to Validity

*Internal Threats.* Performance was highly dependent on instruction; for example, Gpt-5.2 improved from 15.00 % to 81.00 % when prompted to track exceptions. Instruction-tuned models may follow prompts too literally, often failing to predict exceptions because the original prompt did not mention them. Both examples illustrate how the exact choice of prompt can significantly affect outcomes due to *prompt sensitivity*. Additionally, the choice of *model configuration* and sampling hyperparameters may affect results. While our usage of type-aware mutations aims to create a dense neighborhood of local inputs for each program, the approach may introduce latent *distributional biases* in perturbed inputs *e.g.*, longer input sequences, that have a causal relationship with model outcomes. Categorical accuracy for rare exceptions, such as *NameError* or *LookupError*, is unreliable due to *sample sparsity*. Finally, the use of regular expressions, heuristics, and their specific selection & implementations for extracting model responses may introduce *measurement error* or misinterpret the model's intended output.

*External Threats.* While Crux​Eval is currently the standard academic benchmark for code-reasoning output prediction, it consists of small, synthetically generated, standalone Python functions that are a coarse-grained proxy for program execution, potentially masking the nuances of state management in larger systems. Thus, to reduce *benchmark bias*, our findings could be strengthened by applying our approach to multiple, diverse, real-world datasets in future work. By focusing our study on Python due to the choice of benchmark, we acknowledge a *language bias*. Challenges such as predicting *TypeErrors* may not apply to statically-typed languages, and model accuracy & robustness might differ on other languages under the same perturbations. The use of proprietary, black-box models such as Gpt-5.2 or Gemini-3-Pro is tied to specific API versions and may change over time under *model decay*. Our findings are constrained by our perturbation methodology. Without a comprehensive study of perturbations, especially with respect to MPTs, our results may not necessarily generalize to other transformations.

## 6    Related Work

*Robustness and Model Stability on Code.* Early research evaluated code models against semantic-preserving transformations. Henkel et al. [22] and Dong et al. [15] used metamorphic testing (MPT) and "litmus" transformations to assess and augment model stability on code-captioning and classification tasks, respectively. ReCode [45] studied robustness in code *generation* under docstring and code perturbation. Wei et al. [48] applied coverage-guided fuzzing to study the reliability of an LLM determining the semantic equivalence of programs. While benchmarks like EvalPlus [32] provide comprehensive evaluation for generation, our work focuses on the robustness of understanding *given* code.

Recent works have studied deeper semantic perturbations: Hooda et al. [23] used counterfactual mutations, such as flipping branches, to test conceptual understanding, finding significant sensitivity to control-flow changes. Others have examined specific contexts such as poor readability [24], syntactic adversarial attacks in translation [55], and prompt instability in vulnerability detection [20].

Another line of work has sought to tackle the data contamination problem by utilizing dynamic benchmarking, *i.e.*, automatic augmentation and generation of code benchmarks, for both code reasoning [19] and generation tasks [12, 25, 11].

Lam et al. [30] study code reasoning robustness only under *code* perturbation *e.g.*, MPTs, insertion of misleading natural language comments. Our RQ1 is a partial reproduction of their core results; our RQ1 findings broadly align with and reinforce theirs. We generalize the application of perturbation to inputs. *Input perturbation* gives rise to both additional valid, and invalid inputs (which may cause exceptions), allowing us to evaluate dependency on control flow decisions; our RQ2 and RQ3 explore these issues in detail.

Broader exploration of these topics, can be found in surveys by Asgari et al. [3] on metamorphic testing for code models, and Song et al. [41] on the robustness and reasoning failures of LLMs.

*Benchmarking Code Reasoning and Execution.* A significant body of work benchmarks the deeper code-reasoning capabilities of LLMs. We refer to Ceka et al. [8] for a recent survey. Broad multi-task suites like CodeMMLU [34] and SX-Bench [52] assess general software principles, while LiveCodeBench-Exec [26], like CruxEval, assesses output prediction ability. CruxEval-X [51] and Code-Sense [38] further extend these evaluations to multilingual and real-world repository settings. In contrast, REval [10] uses code generation benchmark data to assesses prediction of intermediate states during program execution, and model logical reasoning consistency on code-intelligence tasks of increasing difficulty. Finally, specialized tools such as ExerScope [31] isolate runtime behavior and dynamic properties.

Prenner et al. [36] contribute a benchmark that evaluates models *specifically focused* on their ability to predict exceptions in failing programs, with a prompt tuned to find exceptions. We study robustness broadly on the output *and* exception prediction task on programs in CruxEval, finding limitations, notably for type exceptions, and explore prompt engineering to address these problems.

Patel et al. [35] discuss exception prediction by first building a CFG and then instructing an LLM use the CFG to predict execution results. We study the robustness of LLM code understanding in a setting wherein it lacks the support of a separate CFG analyzer.

Bieber et al. [5] contribute a benchmark that evaluates LLMs' ability to statically determine which exceptions *might* be thrown by a program on *any* input, thus a more difficult task; robustness to perturbation was considered in their work.

Beyond execution tracing, research has moved toward functional equivalence and static analysis. While Core [50] evaluates static information flow, EquiBench [47] and ProbeGen [2] test whether models can determine if two programs are semantically identical. Notably, ProbeGen utilizes the model itself to generate inputs that disprove equivalence, enabling semantic clustering. Finally, recent work has addressed model confidence through calibration frameworks [42] to evaluate and improve code reasoning confidence [46].

## 7    Conclusion

Do language models robustly understand the meaning of programs? We explore this question using perturbations of the CruxEval benchmark, which requires models to correctly predict outputs for given program-input pairs. We perturb the *programs* in CruxEval using meaning-preserving transforms, and the *inputs* using type-aware mutations. Some of the perturbed inputs cause programs to throw exceptions, and we add these to our evaluation.

We find that performance on perturbed inputs and perturbed programs is generally a bit lower, notably *substantially* lower for the frontier Gpt-5.2 model. We also find that performance on exceptions using the given prompt in CruxEval is also substantially worse, and report some prompting interventions that provide evidence of improvement. Finally, we also find that these exception-related interventions help all models improve performance on the original benchmarks, but actually *worsen* the performance of Gpt-5.2.

We find that even frontier models show surprising instruction-sensitive brittleness on output prediction. Future research could explore the causes of and remedies for this brittleness.

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
