# OpenReview forum: "How Robustly do LLMs Understand Execution Semantics?"
_ACM.org/AIWare/2026/Conference — AIware 2026_

### Official Review · Reviewer_v1Xv · 2026-03-10

**Rating:** 3
**Confidence:** 3

**Review:**

I think the paper addresses an important and timely question, and the overall empirical design is stronger than a typical single-benchmark report. The use of multiple perturbation settings, the addition of PSR, and the analysis of exception behavior make the study interesting and potentially useful.

-  I have several methodological concerns. 1) the exception results are strongly confounded by prompt design. Under the original prompt, GPT-5.2 performs very poorly on exception cases, but once the prompt explicitly asks the model to track exceptions, performance improves dramatically. To me, this means that at least part of the “models are poor at exception reasoning” story is really a prompt or benchmark artifact rather than a clean model-semantic limitation. 2) the answer-extraction protocol is somewhat lenient: responses with multiple assertions are counted as correct if any extracted assertion matches, and exception prediction relies partly on heuristic substring matching. Those choices can inflate scores or introduce evaluator subjectivity and measurement noise.

- The paper easy to follow at a high level. The RQ1/RQ2/RQ3 organization works well, and the paper generally makes it clear what is being tested and what the intended takeaways are. I found a few internal inconsistencies. On page 5, the text discusses “DeepSeek R1 Distill 14B” and even a “32B variant,” but those models do not appear in the model list in Sec. 3.2 (page 3) or in Table 3 (page 4). In addition, Table 1 (page  3) and Table 3 (page  4) report different C_0 CruxEval scores for the same models without explaining why.


- I do think this counts as original work, but I would describe it as an original empirical extension or synthesis rather than a radically new paradigm. The paper is clearly operating in an already active area, and it is not inventing robustness evaluation for code LLMs from scratch. That said, I still see meaningful novelty in the combination of contributions here: perturbed-input evaluation on CruxEval, explicit inclusion of exception-causing perturbed inputs, the PSR metric, prompt-based diagnosis of exception prediction, and the analysis of dynamic decision counts. Taken together, those elements form a coherent and worthwhile empirical contribution.

- The significance is somewhat limited by external validity. The study goes deep on CruxEval and its derived variants, but it still relies on one benchmark family, and CruxEval itself consists of short, synthetic Python functions rather than more realistic repository-level code. More broadly, because many of the exception cases are induced by the mutation pipeline, it is not yet clear how representative they are of naturally occurring software bugs. For that reason, I see this as an important benchmark-stress-test paper, but not yet a definitive statement about robust code understanding in real software settings.


Pros
- The paper tackles an important question of whether high benchmark accuracy really indicates robust semantic reasoning in code models.
- The PSR metric is valuable and makes a strong conceptual point.
- The evaluation is multi-dimensional, covering input perturbations, code perturbations, exception cases, and execution decisions.
- The prompt ablation for exception prediction is informative.

Cons
- The headline exception results are strongly confounded by prompt wording; the very large jump from the original prompt to the exception-aware prompt means the paper should be more careful about attributing baseline failure to model-semantic weakness alone.
- The answer-extraction protocol is somewhat weak.
- The broader framing sometimes overreaches. The title, abstract, and conclusion suggest claims about “understanding execution semantics” and even “internal world models vs. pattern matching,” but the actual evidence comes from a single proxy task on short Python functions.
- There are internal inconsistencies as mentioned above.
- The study is limited as it only used one benchmark family, short synthetic Python programs.

**Summary:**

The paper studies LLM code understanding through program-output prediction on CruxEval, treating that task as a proxy for semantic understanding. It organizes the study around three research questions: robustness to input perturbations (RQ1), robustness to meaning-preserving code perturbations such as syntax transformations and variable renaming (RQ2), and robustness as a function of the number of dynamic execution decisions encountered during a run (RQ3). The authors also introduce a new metric, Program-level Strict Robustness (PSR), meant to capture whether a model gets all tested inputs for a program correct. They evaluate 14 models, including open-weight models and three proprietary models.

---

> ### Author Response · Authors · 2026-03-19
>
> Thank you for your thoughtful comments and constructive feedback. We have addressed novelty concerns, and the limitation of Python-only experiments, in a general comment addressed to all reviewers. We would like to address some of the detailed comments:
>
> * Confounded by prompt wording: We have set out to study the brittleness of LLM execution reasoning.  Brittleness includes prompt sensitivity.  Our finding that Gemini’s exception reasoning establishes its brittleness. We did not mean to imply that it entails a “model-semantic limitation” (v1Xv) of LLMs; to the contrary, we ourselves report that a terse and obvious prompt tweak restores its performance on this task.  Nonetheless, execution reasoning, which we observe necessarily includes exceptions, is a goal of the training of frontier models, yet this brittleness crept into the model.  Thus, we maintain that our finding is interesting and important and indicates that such brittleness may lurk in other reasoning tasks.  We will reframe our discussion of this finding to clarify its focus on reasoning brittleness, and not that we have uncovered a fundamental limitation of LLM execution reasoning.
> * We will clarify the motivations for our answer-extraction protocol. During our initial evaluation using the reference CruxEval evaluation harness, we found reasoning models (R1 and GPT 4o families) did not meet performance expectations. After manual review, we found that they did not adhere well to the reference CruxEval prompt that was designed for autoregressive code completion, not the ‘chat-friendly’ style they were trained to follow. We found that models often completed the assertion correctly, but followed it with a user-friendly explanation, often including other example assertions (e.g., counter examples). The CruxEval harness could not mark such a response, despite its arguably useful nature, as correct. To avoid changing the prompt, and thus introducing another prompt choice factor, we relaxed the evaluation mechanism. We still require the LHS of any extracted assertion to match the reference, eliminating the possibility of a false positive wherein output is predicted for a _different_ input. Regarding evaluation of exception prediction, we report a strict match criteria that acts as a lower-bound on performance.
> * We will address our framing to reduce the scope of claims.
> * We will correct the accidental inclusion of R1 14B & 32B in Section 4.2 (RQ2). We intended to report results for these models for RQ1, but experiments did not complete in time. Regarding the Table 1 & 3 discrepancies, we will clarify the reasons for these discrepancies and briefly discuss their scale. We clarify that the program sets are properly overlapping, but not equal, resulting in minor differences. Under RQ1, some programs did not have 10 perturbed inputs and thus were excluded (see Table 1 caption), while under RQ2, 88 programs were excluded as they did not have valid MPT counterparts. To keep the results in each table consistent and interpretable, we report the results on the intersection of program sets, e.g. the same exact n programs before and after MPT.

---

### Official Review · Reviewer_trNk · 2026-03-10

**Rating:** 2
**Confidence:** 4

**Review:**

Pros
+ Addresses an important and timely question on whether LLMs truly understand code execution semantics rather than relying on patterns
+ Introduces a useful program-level strict robustness metric
+ Evaluates robustness along three dimensions

Cons
- The experiments are limited to one programming language, i.e., python, so the findings may not generalize to other languages, particularly statically typed ones.
- The conclusions about model robustness are sensitive to prompting, which raises questions about whether the results reflect the models’ true understanding or the influence of prompt design.
- Some methodological details require further explanation
- Lack of findings and insights

Detailed Comments to Authors:

Novelty

- The paper addresses an important question regarding whether LLMs truly understand code semantics or simply generate outputs based on learned patterns. Although several existing benchmarks report high accuracy for LLMs on code-related tasks, these results do not necessarily demonstrate that the models genuinely understand program semantics. The authors attempt to address this issue by evaluating robustness across three dimensions, which provides an interesting perspective for assessing semantic understanding. However, similar robustness studies have already been explored in prior work, particularly in the context of semantics-preserving code transformations. For example, studies such as “Are Large Language Models Robust in Understanding Code Against Semantics-Preserving Mutations?” have investigated related questions. Therefore, the novelty of this work would be clearer if the authors more explicitly discussed how their approach differs from or extends these existing studies, as well as the limitations of prior methods that their work aims to address.

- The paper introduces the PSR metric as a way to measure whether a model can correctly predict outputs across all inputs for a given program. While this metric is meaningful for SE tasks that naturally involve multiple inputs, its applicability to other code understanding settings is less clear. For example, tasks such as exception prediction may depend primarily on reasoning over the code snippet itself rather than robustness across a set of inputs. In such cases, the paper does not sufficiently explain how PSR should be interpreted or extended. Moreover, PSR appears closely related to simply measuring whether a model passes all test cases associated with a program, so its distinction from existing program-level accuracy notions needs to be better motivated.

- More broadly, many of the techniques used in this study, including input mutation and semantics-preserving code transformations, have already been widely explored in the literature. Prior adversarial robustness work on code models has similarly used such transformations to reveal that models can be misled despite preserving the program semantics, which overlaps substantially with the motivation behind RQ2. Even simple transformations such as variable renaming can cause models to produce different outputs compared to the original program, an observation that has already been widely reported in prior literature. As a result, the novelty of this part of the paper would be stronger if the authors clarified how their evaluation meaningfully differs from or extends those earlier robustness studies.

- In addition, the study is entirely based on the CruxEval benchmark, which focuses only on Python programs. While starting with simple tasks is a reasonable first step, this design choice limits the broader novelty and generalizability of the work. It remains unclear whether the observed robustness patterns would hold for other programming languages.

Methodology Soundness

- The evaluation across the three RQs appears generally sound and aligned with the goals of the study. However, several aspects of the experimental setup require further clarification to properly assess the rigor of the methodology.

- First, the paper would benefit from a brief overview of the CruxEval benchmark, including the number of tasks, the nature of the output prediction task, and the number of transformations or perturbations applied. Providing these details would help readers better understand the scale of the experiments and assess the completeness of the evaluation.

- Regarding RQ1, I have two concerns about the experimental design. The authors apply a type-aware mutation strategy following prior work [28] (lines 169–197). However, it is not clear whether the authors verified that the generated mutations were valid and applicable to the corresponding programs. In practice, some mutations may produce invalid inputs or lead to cases where a correct program cannot produce a valid output. Therefore, it would be important for the authors to clarify whether mutation correctness was verified and how invalid mutations were filtered.

- My second concern relates to the use of an execution time limit of 5 seconds to identify potential infinite loops or recursion. Exceeding this threshold does not necessarily imply that the code has entered an infinite loop, as some valid programs may simply take longer to execute. The paper would benefit from reporting how many programs actually exceeded this limit and whether manual verification was performed to confirm that these cases correspond to genuine infinite loops. More fundamentally, the motivation for enforcing a time limit in this study is somewhat unclear. Since the primary goal is to evaluate whether LLMs understand code semantics by predicting correct outputs, the correctness of the predicted behavior should arguably be the main concern rather than execution time. In principle, a model could reason correctly about a program even if the execution takes longer than the chosen threshold. Clarifying the role of the time limit in the evaluation would help strengthen the methodological justification.

- For RQ2, additional details are needed regarding how the program transformations were applied. For example, it would be helpful to clarify whether multiple transformations were applied sequentially to the same program or whether each transformed variant contains only a single transformation. This information is important for interpreting the robustness results and for enabling fair comparison with related studies.

- Finally, I appreciate that the authors evaluate a diverse set of models, including several open-source small-scale models (1.5B and 7B parameters) in addition to frontier models. This design choice is valuable because it helps analyze the inherent limitations of smaller models while avoiding the high cost of running large proprietary models for extensive experiments. However, I would like to understand the rationale for including math-specialized models such as Qwen2.5-Math (1.5B and 7B) in a study focused on code reasoning. It would be helpful if the authors clarified why these models were chosen instead of code-specialized variants such as Qwen2.5-Coder, which may be more directly aligned with the programming tasks studied in this paper.

Significance

- The results reported for the three research questions are interesting, particularly the findings related to the discrepancy in model performance when predicting exceptions. However, the study would benefit from providing deeper insights and clearer takeaways beyond the reported statistics. While the tables and quantitative results provide useful evidence, the paper currently presents relatively limited discussion on what these findings imply for the broader development of LLMs in software engineering tasks.

- For example, the observation regarding the potential “sycophantic” behavior of models discussed at the end of Section 4.1 is particularly interesting. However, this insight appears only briefly and would benefit from further elaboration. Providing concrete examples of cases where models fail under different perturbations or transformations could significantly improve the interpretability of the results. Additionally, it would be valuable to analyze relationships across the three robustness dimensions studied in the paper and explain which types of programs or transformations tend to cause failures.

- More generally, the results are currently presented in a largely statistical manner. While statistical validation is important, it would be helpful if the authors used the quantitative results as a foundation to derive stronger qualitative insights. For instance, including illustrative examples, discussing patterns of model failures, and explaining why certain models behave differently would make the findings more informative to the research community.

- Furthermore, the paper would benefit from a clearer discussion of the implications for future work and real-world software engineering practice. For example, the results raise interesting questions about whether current LLM architectures are sufficient for reasoning about program semantics, or whether new training strategies and model designs are required. The authors could also discuss whether improving semantic understanding requires architectural changes, specialized training data, or new evaluation frameworks.

**Summary:**

Current LLMs demonstrate strong performance in many SE tasks, but it remains unclear whether this performance reflects genuine understanding of code semantics. Although existing benchmarks report high accuracies, evaluating the robustness of these models is essential to determine their true reasoning capability. This paper aims to improve the evaluation of LLM code understanding by using program output prediction as a proxy for semantic comprehension. To achieve this, the authors assess the robustness of LLMs across three key dimensions: predicting program outputs under input perturbations, under semantic-preserving program transformations, and across execution traces with varying numbers of decisions. These evaluations are conducted using the CruxEval benchmark on 14 models.

---

> ### Author Response · Authors · 2026-03-19
>
> Thank you for your thoughtful comments and constructive feedback. We have addressed novelty concerns, and the limitation of Python-only experiments, in a general comment addressed to all reviewers. We would like to address some of the detailed comments below:
>
> Methodology Soundness
>
> * We will add a brief overview of the CruxEval benchmark.
> * We discuss verification of generated mutations under Section 3.1 RQ1. We execute all mutated inputs with the reference program. If a generated input mutation did not produce a runtime exception, the input is valid. If an exception was raised, we record the input for the CruxEval_exc dataset, and expect the LLM to correctly predict the exception.
> * We will clarify the 5 second execution limit. The limit is only applicable during the mutation (perturbation) process, not the _evaluation_ process. During evaluation, the model’s predicted output e.g., `[1,2,3]` is compared to the reference output e.g., `[1,3,2]`, by producing an equality expression `[1,2,3] == [1,3,2]`, and evaluating this expression’s truth value using a common expression evaluator [`asteval`](https://github.com/lmfit/asteval) (210 GitHub stars, 58M downloads). In contrast, the reference CruxEval [evaluation harness](https://github.com/facebookresearch/cruxeval/blob/190faf16d175b5847b0af05d937872b1fb395942/evaluation/utils_execute.py#L16) has a 3 second execution limit, and during evaluation, executes the entire program-input-equality chain for each generation. We chose 5 seconds during our perturbation phase to limit the risk of false negatives i.e., valid mutated input incorrectly marked as invalid during validation, by relaxing the time constraint to account for potential instrumentation overhead and local, limited-resource execution. We did not increase this limit further as it only applies during the validation of input mutations (perturbation step), requiring repeated mutate-execute loops. To address the concern, we have rerun (executed method with input) all $\sim8.5$k program-input pairs (original & perturbed), and observed a maximum execution time of 7.95 ms with overhead (forking, instrumentation, etc.), and 976 µs for just the program-input itself (excluding overhead). This suggests that the execution time limit has limited to no effect on our (non-exception) results. We also reran CruxEval_exc with the 5s limit, and a doubled **10s** limit, and observed a maximum execution time of 4.09 ms (w/ overhead) for non-TimeoutError cases, and found no difference in results on TimeoutError: the same 36 (see Table 5) TimeoutError cases persisted with the doubled time limit. While these are not rigorous experiments, it suggests 1) TimeoutError cases are unlikely to be borderline 2) the order of magnitude difference in execution times ($\sim8$ ms vs 10 s) suggests any program-input pairs that _could_ complete given more time, are extreme outliers. Thus, we consider the multi-second execution limit as generous given limited program-input complexity, and having limited to no impact on our evaluation. We will perform a manual verification of the 36 TimeoutError cases to identify genuine infinite loops, and add a brief discussion of the execution limit, empirical runtimes, and clarification of how & where the execution limit applies.
> * We will clarify that under RQ2, only a single transformation was applied per program.
> * We will clarify the motivations for our model choice. DeepSeek R1 Distill models are fine-tunes of open-source models using generations from R1. We included Qwen2.5-Math (1.5B and 7B) as they are the explicit ancestors (i.e., starting checkpoints) of DeepSeek R1 Distill (1.5B and 7B). See DeepSeek [model card](https://huggingface.co/deepseek-ai/DeepSeek-R1-Distill-Qwen-7B#deepseek-r1-distill-models). This allows us to isolate the effect of GRPO reasoning training over prior training approaches used in Qwen2.5-Math.
> * We will address other concerns, space permitting.

---

### Official Review · Reviewer_FJWD · 2026-03-11

**Rating:** 3
**Confidence:** 5

**Review:**

## Strengths

1. Important Research Question. The paper addresses a highly relevant problem in the evaluation of LLMs for software engineering tasks: whether models genuinely understand program semantics.


2. Clear Experimental Design. The three research questions (input perturbations, code transformations, and exception prediction) provide a clear structure for analyzing robustness across multiple dimensions of program execution understanding.

3. Insightful Empirical Findings. The paper reveals an interesting contrast between frontier models with high accuracy but brittle robustness and open-source reasoning models with lower accuracy but greater stability, which contributes valuable insights into the behavior of current LLM architectures.


##  Weaknesses

1. Limited Novelty in Evaluation Paradigm. While the empirical results are interesting, the overall evaluation approach—testing robustness via perturbations and semantic-preserving transformations—is conceptually similar to prior work [1, 2, 3, 4] on robustness testing for code models. The paper could better clarify how its methodology significantly differs from or improves upon existing robustness studies.

2. Dependence on a Single Benchmark. Most experiments rely heavily on CruxEval. Although the benchmark is suitable for output prediction tasks, relying on a single dataset may limit the generalizability of the conclusions. Evaluating on othe benchmarks such as CodeSense [5] or CoRe[6] would strength the claim.

3. Limited Analysis of Failure Causes. While the paper identifies robustness issues, the analysis of why models fail remains somewhat shallow. For example, it is unclear whether failures arise from: reasoning limitations, token-level pattern matching, prompt sensitivity, or overfitting. A deeper investigation into these causes would improve the scientific contribution.

4. Insufficient Details on Some Experimental Settings. Some implementation and evaluation details could be clearer, such as sampling parameters and decoding strategies.

5. Missing related work: several studies on dynamically generating programming problems for benchmarking code LLMs are not discussed.

[1] DyCodeEval: Dynamic Benchmarking of Reasoning Capabilities in Code Large Language Models Under Data Contamination. ICML 2025

[2] DynaCode: A Dynamic Complexity-Aware Code Benchmark for Evaluating Large Language Models in Code Generation. ACL 2025 Findings

[3] Is your benchmark (still) useful? dynamic benchmarking for code language models. Arivx

[4] PPM: Automated Generation of Diverse Programming Problems for Benchmarking Code Generation Models. FSE 2024.

[5] CodeSense: A Real-World Benchmark and Dataset for Code Semantic Reasoning.

[6] CoRe: Benchmarking LLMs' Code Reasoning Capabilities through Static Analysis Tasks

**Summary:**

This paper investigates whether large language models (LLMs) truly understand program execution semantics or instead rely on superficial pattern matching. The authors study this question through the lens of robustness in program output prediction, a task that requires models to predict the exact output of a program for a given input. The study evaluates several LLMs on CruxEval, a benchmark designed to assess semantic code understanding. The authors introduce robustness tests by applying multiple forms of semantics-preserving program transformations and input perturbations. These transformations do not change the correct output but alter the surface form of the program or the characteristics of the input, allowing the authors to test whether models truly reason about execution semantics.
The results reveal notable differences across models. Frontier models (e.g., GPT-5.2) achieve nearly perfect accuracy on the original benchmark but experience significant performance drops under perturbations, indicating brittle behavior. In contrast, some open-source reasoning models (e.g., DeepSeek-R1 family) exhibit lower overall accuracy but more stable performance under perturbations. The authors also analyze prediction failures involving exceptions and explore several mitigation strategies to improve exception prediction.

---

> ### Author Response · Authors · 2026-03-19
>
> Thank you for your thoughtful comments and constructive feedback. We have addressed novelty concerns, and the limitation of Python-only experiments, in a general comment addressed to all reviewers. We would like to address the remaining concerns below:
>
> Limited Analysis
> * We will improve failure analysis by performing a case study on a sample of incorrect model generations and reporting the results in Section 5 Discussion. Large scale automated analysis of model failure is a promising direction for future work.
>
> Experimental Settings
> * We will clarify implementation and evaluation details under Section 3.2, e.g., ```We utilized HuggingFace `text-generation-inference:3.3.6` (tgi) for inference of local models on 2 x TITAN RTX GPUs. We specified `do_sample=False` i.e., greedy decoding, and a maximum generation length of 16,384. To follow developer workflows using frontier models [...]```
>
> Related Work
> * We will discuss related work [1, 2, 3, 4] on dynamic programming problem generation in Section 6, positioning our work as a method for augmenting _existing_ code understanding benchmarks with _input_ perturbations for the purpose of evaluating _robustness_. In contrast, prior work has focused on synthesizing new problems, limiting their value for understanding how LLMs function on specific problems, _given_ varied inputs.

---

> > ### Comment · Reviewer_FJWD · 2026-03-19
> > **Response to Authors**
> >
> > Thanks for the authors response, which addressed most of my concern, however, the evaluation on a single benchmark still limit the generalizeability of this work, so I will keep my score as weak accept.

---

### Author Response · Authors · 2026-03-19
**General Comment to All Reviewers**

We sincerely thank all reviewers for their time, thoughtful comments, and constructive feedback. We are encouraged by the positive feedback regarding the importance of our research questions. We address some common areas of concern below:

Single benchmark & language

* Our methodology centers on Python and CruxEval to ensure both industry relevance and benchmarking consistency. While newer benchmarks like LiveCodeBench-Exec have since emerged, CruxEval provided the necessary stability for our specific experimental pivot: testing the robustness of the 'single input-output' constraint. By isolating this variable within the most recognized code execution reasoning benchmark, we have uncovered a potential brittleness in current LLM execution reasoning that sheds new light on prior results.


Novelty

* The question of understanding LLM reasoning is central to effectively harnessing them to reliably solve problems for society.  Reasoning about execution is a particularly important instance of this question because the executability of code allows experiments to definitely test whether understanding code matches its output. Our field is uniquely positioned to lead this investigation because of our shared expertise in program analysis.  The very importance of the problem justifies its thorough experimental investigation.  We contribute to this effort by employing novel methodologies and a key reproduction.
* Concerning methodology, we argue that code understanding requires more than navigating a single 'happy path'. This observation led us to two methodological novelties in our use of CruxEval. First, we are the first to consider sets of inputs; a model’s ability to predict behavior across a range of values is essential for assessing whether its understanding is robust and establishing confidence in that finding. Second, we are the first to extend the evaluation to type-invalid inputs and exceptions. In addition to these input dimensions, we are the first to consider the robustness of LLM code understanding as a function of the discrete control decisions along a path.
* We have contributed PSR, a new robustness metric. Unlike accuracy measures, PSR focuses on measuring the proportion of a set of programs on which a _model_ completely succeeds on the output prediction task modulo each program’s individual test suite. This is in contrast to the traditional measure of success relative to a single program and its pass rate on its test suite. PSR is a demanding measure, inspired by our insistence that true understanding be complete.
* Finally, a core contribution of our work is the rigorous reproduction of LLM execution reasoning under meaning-preserving transformations (MPT) in RQ2. While our field often prioritizes novelty, the stability of the scientific record depends on independent verification. This reproduction provides a necessary empirical anchor for the field’s understanding of model robustness to MPT.  We will update the discussion of RQ2 to clarify its role as a reproduction.